# Efficacy and Safety of Polyunsaturated Fatty Acids Supplementation in the Treatment of Attention Deficit Hyperactivity Disorder (ADHD) in Children and Adolescents: A Systematic Review and Meta-Analysis of Clinical Trials

**DOI:** 10.3390/nu13041226

**Published:** 2021-04-08

**Authors:** Mina Nicole Händel, Jeanett Friis Rohde, Marie Louise Rimestad, Elisabeth Bandak, Kirsten Birkefoss, Britta Tendal, Sanne Lemcke, Henriette Edemann Callesen

**Affiliations:** 1The Parker Institute, Bispebjerg and Frederiksberg Hospital, The Capital Region, 2000 Frederiksberg, Denmark; jeanett.friis.rohde@regionh.dk (J.F.R.); elisabeth.ann.bandak@regionh.dk (E.B.); 2The Danish Health Authority, 2300 Copenhagen, Denmark; kibi@sst.dk (K.B.); brittatdk@gmail.com (B.T.); henriette.callesen@gmail.com (H.E.C.); 3Institute of Psychology, Aarhus University, 8000 Aarhus, Denmark; psykolog-rimestad@protonmail.com; 4Department of Child and Adolescent Psychiatry, Aarhus University Hospital, Psychiatry, 8200 Aarhus N, Denmark; Sanne.Lemcke@ps.rm.dk

**Keywords:** fatty acids, omega 3, polyunsaturated, attention deficit hyperactivity disorder, ADHD, children, adolescents, systematic review, meta-analysis

## Abstract

Based on epidemiological and animal studies, the rationale for using polyunsaturated fatty acids (PUFAs) as a treatment for Attention Deficit Hyperactivity Disorder (ADHD) seems promising. Here, the objective was to systematically identify and critically assess the evidence from clinical trials. The primary outcome was ADHD core symptoms. The secondary outcomes were behavioral difficulties, quality of life, and side effects. We performed a systematic search in Medline, Embase, Cinahl, PsycInfo, and the Cochrane Library up to June 2020. The overall certainty of evidence was evaluated using Grades of Recommendation, Assessment, Development, and Evaluation (GRADE). We identified 31 relevant randomized controlled trials including 1755 patients. The results showed no effect on ADHD core symptoms rated by parents (k = 23; SMD: −0.17; 95% CI: −0.32, −0.02) or teachers (k = 10; SMD: −0.06; 95% CI: −0.31, 0.19). There was no effect on behavioral difficulties, rated by parents (k = 7; SMD: −0.02; 95% CI: −0.17, 0.14) or teachers (k = 5; SMD: −0.04; 95% CI: −0.35, 0.26). There was no effect on quality of life (SMD: 0.01; 95% CI: −0.29, 0.31). PUFA did not increase the occurrence of side effects. For now, there seems to be no benefit of PUFA in ADHD treatment; however, the certainty of evidence is questionable, and thus no conclusive guidance can be made. The protocol is registered in PROSPERO ID: CRD42020158453.

## 1. Introduction

Attention Deficit Hyperactivity Disorder (ADHD) is a common neurodevelopmental disorder in children and adolescents, which may persist into adulthood. A meta-analysis of 102 prevalence studies found that the worldwide prevalence estimate for ADHD among children and adolescents under the age of 18 years is 3.4% (CI 95% 2.6–4.5) [1].

ADHD is characterized by three core symptoms, namely, inattention, hyperactivity, and impulsivity. The symptoms must be present in different settings and be both impairing and age inappropriate. ADHD is frequently comorbid with other psychiatric disorders and a substantial burden to the affected children and their families [2]. Long-term studies have revealed that a diagnosis of ADHD is associated with lower educational achievements and significant higher prevalence of, e.g., injury, substance abuse, unemployment, and delinquency [3,4,5,6,7,8].

A wide variety of treatments such as pharmacological and psychosocial interventions are used for the management of ADHD. There is some evidence for the short-term effectiveness of certain non-pharmacological interventions and pharmacological treatments [9,10,11]. However, some patients and parents are concerned with the use of pharmaceuticals for the treatment of ADHD, and initiation of pharmacological treatment may be linked with a certain degree of reluctance due to side-effects. Investigations of other interventions such as diet or nutritional supplements are therefore necessary.

Based on evidence from epidemiological studies and animal models, the rationale for using polyunsaturated fatty acids (PUFAs) as a treatment option for ADHD is promising. Studies have found that a deficiency of essential fatty acids positively correlates with ADHD symptoms [12]. Supporting the evidence that PUFAs may play a role in brain disorders, essential fatty acids have shown to regulate neurotransmitter and immune functions via the modulation of lipid rafts signaling platforms on the cell membrane in addition to having an anti-inflammatory inhibition of the free radical generation and oxidative stress [13,14,15].

The objective of this systematic review and meta-analysis was to systematically identify and critically assess the current evidence from clinical trials concerning the administration of supplementation with PUFAs for the treatment of ADHD among children and adolescents (6–18 years). Specifically, we sought to evaluate the effect of supplementation with PUFAs on ADHD core symptoms and behavioral difficulties, rated by both parent and teachers. Furthermore, we sought to investigate the impact on quality of life as well as the occurrence of side effects including diarrhea, gastrointestinal discomfort, and nausea.

The systematic review and meta-analysis were based on an update of the results from a Danish national clinical guideline published in 2018 by the Danish Health Authority.

## 2. Materials and Methods

Established methods recommended by the Cochrane Collaboration [16] and according to the principles described in the Grades of Recommendation, Assessment, Development, and Evaluation (GRADE) [17] approach were used to conduct this systematic review and meta-analysis. Moreover, the systematic review was structured according to Population, Intervention, Comparison, and Outcome (PICO) characterization [18]. Specifically, the PICO question was whether children and adolescents aged 6–18 years with ADHD should be offered PUFA supplementation. The definitions of population, intervention, comparator, outcomes, and study design are specified in Table 1. The research question is best addressed through a randomized study design in order to evaluate the effectiveness of the intervention so that issues regarding both measured and unmeasured confounding are minimized.

The protocol was registered in the International Prospective Register of Systematic Reviews (PROSPERO: CRD42020158453 (accepted 28 April 2020) and the systematic review was constructed in accordance to the Preferred Reporting Items for Systematic review and Meta-Analysis (PRISMA) statement [22,23] (PRISMA checklist is provided in the Appendix A).

### 2.1. Search Strategy

As a part of the conduction of the Danish national clinical guideline on ADHD by the Danish Health Authority, a literature search were performed by KB in September 2013 (with no restriction on date), in November 2017 (dates restricted to 2013–2017), and again in June 2020 (dates restricted to 2017–2020). The systematic literature search was performed in multiple databases including Medline, Embase, Cinahl, and PsycInfo and limited to randomized controlled trials. The original search strategy included a search for both systematic reviews and individual randomized controlled trials. The justification for the stepwise search strategy is that the Danish Health Authority prioritize to enrich existing guidelines with all available eligible literature approximately every third year. All search strategies for the 2013, 2017, and 2020 searches included medical subject heading (MeSH) and text words related to our eligibility criteria. As an example MeSH terms for the intervention were “unsaturated fatty acid” or “Diet therapy” or “diet supplementation” or “Fish oil” or “Carnitine “, or text words for the intervention ((fatty adj1 acid*) or ((Polyunsaturated or poly-unsaturated or unsaturated) adj1 (fat or fatty)) or omega-3 or omega3 or omega 3 or omega-6 or omega6 or omega 6 or (docosahexaenoic adj acid*) or (eicosapentaenoic adj acid*) or (arachidonic adj acid)) or ((fish adj1 oil*) or cod liver oil* or lax oil* or tuna oil* or carnitine or Levocarnitine or “L Carnitine” or L-carnitine or bicarnitine) or ((diet* or food or nutrition) adj1 (therapy or supplement*)) (for further details please see the example presented in Table 2 and in the search protocols provided in Appendix A). Both the original and updated searches were limited to literature written in English, Danish, Norwegian, and Swedish due to limitations in language proficiency in the author group. Moreover, to ensure that any relevant studies were not missed by the search, content experts from the guideline working group were conferred with, and reference lists of included articles and previous reviews were screened for potentially relevant studies. Conference abstracts were considered if data were not published elsewhere. Study authors were not contacted to identify additional studies.

### 2.2. Study Selection

The search results were imported to RefWorks (Refworks (online software) https://www.refworks.com, accessed on 2 March 2021), and duplicate references were removed. Subsequently, the records were imported into Covidence software (Covidence (Online software)). Covidence team https://www.covidence.org/home, accessed on 2 March 2021) was used for the screening process and data management. 

The titles and abstracts of the identified references were screened by one reviewer (MLR), to assess according to the inclusion criteria (Table 1). Subsequently, the full text of potential studies was independently screened by two review authors out of a group of review authors (MLR, SL, HC, JFR, MNH) for eligibility. Disagreement was discussed or by consultation of a third reviewer (BT). Neither of the review authors were blinded to the journal titles, study authors/institutions, or year of publication. 

### 2.3. Extraction of Individual Randomized Controlled Studies

Data extraction was conducted using a predefined template in Covidence software. The assessment of the studies included study settings, population demographics and baseline characteristics, details on intervention and control conditions, study design, outcome, and time of measurement. Two review authors out of a group of review authors (MLR, SL, HC, EB, JFR, MNH) independently extracted data in duplicate. Discrepancies were resolved through discussion in the review team.

Data for analysis were also extracted in Covidence (Online software: https://www.covidence.org/home, accessed on 2 March 2021) and afterwards exported to Review Manager (version 5.2) (Review Manager (RevMan) (Computer program)) (2014). None of the study authors were contacted by email to provide additional information to resolve uncertainties or to obtain missing data. If multiple reports of a single study were identified, the publication with most complete data was included.

### 2.4. Quality Assessment

Quality assessment of the evidence was evaluated using GRADE method [24]. There are four possible ratings of the quality: high, moderate, low, and very low. Downgrading was carried out for each outcome using the standard definition of risk of bias, inconsistency, indirectness, imprecision, and publication. The overall quality of evidence was based on the lowest quality of the primary outcome and reflected the extent to which we are confident that the effect estimates are correct.

Two review authors out of a group of review authors (MLR, SL, HC, EB, JFR, MNH) independently assessed risk of bias of the primary studies by using the Cochrane risk of bias tool [20] that includes the following characteristics: randomization sequence generation, reatment allocation concealment, blinding of patients and personnel, blinding of outcome assessors, completeness of outcome data, selective outcome reporting, other sources of bias. Discrepancies were resolved through discussion.

### 2.5. Meta-Analysis

For a dichotomous outcome, the relative risk (RR) was calculated, including the 95% confidence interval (CI). When few or zero events occurred in both the intervention and control group, a risk difference (RD) analysis was performed as sensitivity analysis. For a continuous outcome, since different measurement scales were used, we calculated effect size using the standardized mean difference (SMD), including the 95% confidence interval. Random effect models were used to calculate pooled estimates of effects. Furthermore, a funnel plot was produced to assess the publication bias across the studies, when more than 10 studies were included. Statistical heterogeneity was quantified using I^2^ statistic [25], with an I^2^ value greater than of 50% considered to be substantial heterogeneity. 

Review Manager Software (version 5.3) (The Nordic Cochrane Centre, The Cochrane Collaboration, Copenhagen, Denmark) was used to produce the analyses and forest plots.

## 3. Results

### 3.1. Literature Search 

In the 2013 search, one systematic review article was found [9], from which 12 randomized trials were identified among the included studies [26,27,28,29,30,31,32,33,34,35,36,37]. These were supplemented by 11 randomized trials from an updated literature search in 2017 [38,39,40,41,42,43,44,45,46,47,48] and a further eight randomized studies from the latest updated search in 2020 [49,50,51,52,53,54,55,56].

Thus, the evidence base consists of a total of 31 randomized trials.

A flowchart of included secondary and primary studies is presented in Figure 1; Figure 2, and a list of excluded studies after full-text screening including reasons for their exclusion is presented in Appendix A.

### 3.2. Review of the Evidence

The populations in the included studies consisted of children with ADHD in the age group ranging from 6–18 years. The interventions consisted of supplements with polyunsaturated fatty acids with either omega 3, omega 6, or combined supplements with both types of fatty acids. The interventions lasted between 8 weeks to 12 months. In some studies [38,49,55], the children were also in medical treatment in both the intervention group and the placebo group, whereby fatty acid treatment was investigated as an active add-on treatment in the intervention group. Information on study identification, baseline characteristics, intervention, control, reported outcomes, and the authors’ conclusion is presented in Appendix A.

#### 3.2.1. Primary Outcomes

The primary outcomes were reported in 24 studies [29,32,34,35,37,38,39,40,41,42,44,45,46,47,48,49,50,51,52,53,54,55,56] and were based on information from 1755 participants. The results showed no clinically relevant effect was found on the primary outcome parent-rated ADHD core symptoms (SMD: −0.17; 95% CI: −0.32, −0.02) (Figure 3), corresponding to a mean difference on the Parent ADHD rating scale of −1.85 (95% CI: −3.49, −0.22) calculated from the endpoint SD in the control group of Cornu et al., 2018 [51]. The Parent ADHD rating scale ranges from 0 to 54, and thus the result is equivalent to a decrease of 3.4% on the scale (MCID is estimated to a decrease of 30% [19,20,21]), nor was there any effect found on teacher-rated ADHD core symptoms (k = 10; SMD: −0.06; 95% CI: −0.31, 0.19) [27,29,32,33,35,41,45,50,53] (Figure 4).

#### 3.2.2. Secondary Outcomes

There was no clinically significant effect on behavioral difficulties, neither when the outcomes were rated by parents (k = 7; SMD: −0.02; 95% CI: −0.17, 0.14) [29,32,35,44,50,52,54] (Figure 5) nor by teachers (k = 5; SMD: −0.04; 95% CI: −0.35, 0.26) [29,32,35,39,50] (Figure 6).

Only two studies [32,52] examined the effect of the intervention on quality of life, with no significant effect (SMD: 0.01; 95% CI: −0.29, 0.31) (Figure 7).

Additionally, there were no differences between the intervention group compared to the placebo group regarding the three selected side effects: diarrhea (k = 5, RR: 1.08; 95% CI: 0.32, 3.63 and RD: −0.00; 95% CI: −0.04, 0.03) [29,42,49,50,55], gastrointestinal discomfort (k = 4; RR: 0.72; 95% CI: 0.27, 1.88 and RD: −0.01; 95% CI: −0.04, 0.03) [42,43,44,55], and nausea (k = 6, RR: 0.99; 95% CI: 0.41, 2.38 and RD: 0.01; 95% CI: −0.02, 0.03) [29,32,33,42,43,49,55] (Figure 8).

### 3.3. Certainty of Evidence (GRADE)

The certainty of the evidence was overall low to very low, as there were problems with inaccurate effect estimates in small studies (very serious risk of imprecision) as well as serious risk of bias. The risk of bias was primarily due to problems with blinding of the observers that assessed the effect (parent and teachers). However, the studies generally suffered from methodological flaws and were largely poorly described, especially regarding how the randomization sequence was generated, how the allocation was concealed, and how incomplete data were handled.

## 4. Discussion

The objective of this systematic literature review and meta-analysis was to provide clinicians, caregivers and guideline panels with an updated overview and critical assessment of the evidence, investigating the effect of PUFA among children aged 6 to 18 years diagnosed with ADHD. Based on a substantial body of evidence, there is no indication that supplementation with PUFA has a positive effect on core symptoms of ADHD, behavioral difficulties, or quality of life. The present review suggests that there is no substantial increase in side effects following the use of PUFA, including the occurrence of diarrhea, gastrointestinal discomfort, or nausea.

Based on our findings, there is insufficient evidence to support patients, parents, clinicians, and caregivers in their decision on whether to use PUFA as a supplementation in the treatment of ADHD. Consequently, the patient preferences are expected to be unambiguous, in the sense that some parents will prefer dietary changes rather than pharmacological treatment, and other parents will find it difficult and relatively intrusive to implement dietary changes in children and adolescents. Effective treatment with PUFA would supposedly depend on the presence of an initial PUFA deficiency observed in the patient at baseline. A significant decrease in PUFA levels has indeed been observed in patients with ADHD as compared to healthy controls [57]. However, it still needs to be investigated what role PUFA plays in the pathology of ADHD. It remains unknown whether PUFA deficiency represents a neuropathological finding directly potentiating symptom outbreak, or rather a compensatory mechanism due to long-standing disease manifestation. In addition, an evaluation of the long-term effects of providing PUFA as a supplement is needed. For now, the effect has only been tested in patients over a time period ranging from 8 weeks to 12 months. 

The amount of RCTs investigating the use of PUFA as a treatment for ADHD has increased since the latest Cochrane review by Gilles et al. was published in 2012 [58]. Despite an increase in the number of available studies, the conclusion and quality of the evidence remains unchanged. Thus, the evidence still indicates an inability of PUFA to effectively alter the symptomatology of ADHD. The evidence remains of a low to very low quality, thus reflecting a high degree of uncertainty of the effect estimates. 

The risk of bias in the identified studies includes an inadequate random sequence generation and allocation concealment, which in RCTs is mandatory to ensure that intervention and control groups are kept as identical as possible. Thus, the inability to perform sufficient random allocation may induce systematic errors, which may have a considerable impact on the final results. Other major sources of bias were due to incomplete outcome reporting, thus reflecting an increased risk of reporting bias and attrition. Regarding incomplete data, a review has previously described in which problems with drop-out seem to be common in n-3 long chain PUFA supplementation trials in children and adolescents in general [59]. This indicates that dropouts may be a common, inherent issue when seeking to investigate the effect of PUFA in a research setting. 

Blinding of the observers evaluating the effect is essential. The included studies generally displayed an unclear level of blinding, as blinding in the majority of studies was inadequately described. Blinding of participants may once again constitute an inherent problem when investigating the effect of PUFA due to the distinct smell and taste [60]. This may unmask the allocation to the respective groups, as patients and parents over time may become aware if they indeed are receiving the active PUFA component. Nevertheless, adequate blinding of observers not directly related to the child should be possible, including the researcher and teachers. Collectively, these problems lead to a high risk of bias, which may essentially have an impact on the results and thereby lead to a distortion in the final conclusion. 

Our findings displaying an inadequate effect of PUFA in the treatment of ADHD is in line with many previous reviews on the subject [57,61,62,63,64,65], but not all [66,67,68,69].

When comparing our review to others reporting a beneficial effect, it becomes evident that there is a discrepancy in the methodology used to evaluate treatment effects, which may explain the discrepancy in results. In three of the reviews [66,67,69], no specific analysis to obtain pooled estimate of effects was performed, as results from the individual trials were only narratively described. This prevents a direct comparison between trials, including an assessment of the overall magnitude of effect. Common for all reviews reporting a positive effect was a lack of quality assessment of the included studies. This hinders an evaluation of the extent of trust in the data and may essentially mask some issues that otherwise would lead to a downscaling of the certainty in the evidence, and thereby the confidence in the results. It should be mentioned, apart from ADHD core symptoms, the reviews reporting a positive effect also include other outcomes not evaluated in the present review. The inclusion and exclusion criteria furthermore varied across the reviews. This may in part also explain some of the discrepancy in the reported results. 

As mentioned above, several inherent methodological issues due indeed exist when it comes to investigating the effect of PUFA in research settings. Nevertheless, this does not unconditionally prevent the possibility of performing high-quality research on the matter. As such, future research should focus on conducting clinical trials of high-quality, following the CONSORT (Consolidated Standards of Reporting Trials) Statement [70]. This is essential if we wish to move forward and be able to conclusively evaluate the role of PUFA in the treatment of ADHD.

### Strengths and Limitations of the Current Review

In order to ensure high methodological quality, this systematic review and meta-analyses followed the guidelines of the Cochrane Collaboration and PRISMA as well as pre-registering a protocol at PROSPERO. Moreover, a comprehensive search and duplicate full-text study selection, data extraction, and quality assessment were used. Amongst the limitations, we acknowledge that since the search was limited/restricted to English and Scandinavian languages, there might be relevant studies unidentified. Moreover, grey literature was not searched, and thus relevant studies may have been unidentified. Furthermore, the authors of the included studies were not contacted for further information, and the results are merely based on data published in peer-reviewed articles. The review authors were not blinded in the process of selecting literature. 

## 5. Conclusions

Based on the current low to very low evidence, there seems to be no benefit of providing PUFA supplements to children and adolescents with ADHD concerning parent- or teacher-rated core symptoms, behavioral difficulties, or quality of life. Concerns on adverse effects of PUFA supplementation is limited. Conclusive guidance for patients, parent and clinical practice cannot be made due to the many limitations inherent to the included studies.

## Figures and Tables

**Figure 1 nutrients-13-01226-f001:**
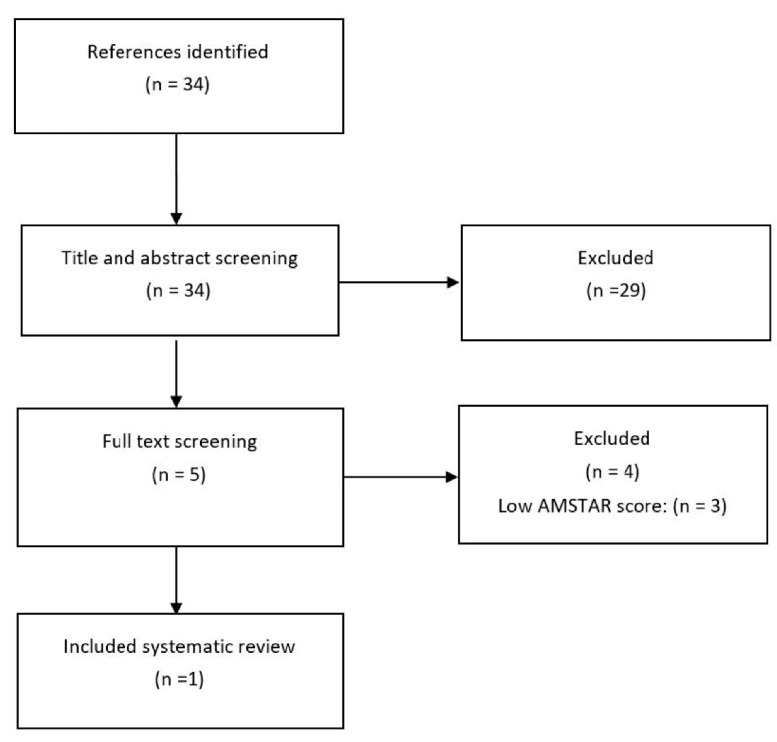
Flowchart of the search for systematic reviews in 2013.

**Figure 2 nutrients-13-01226-f002:**
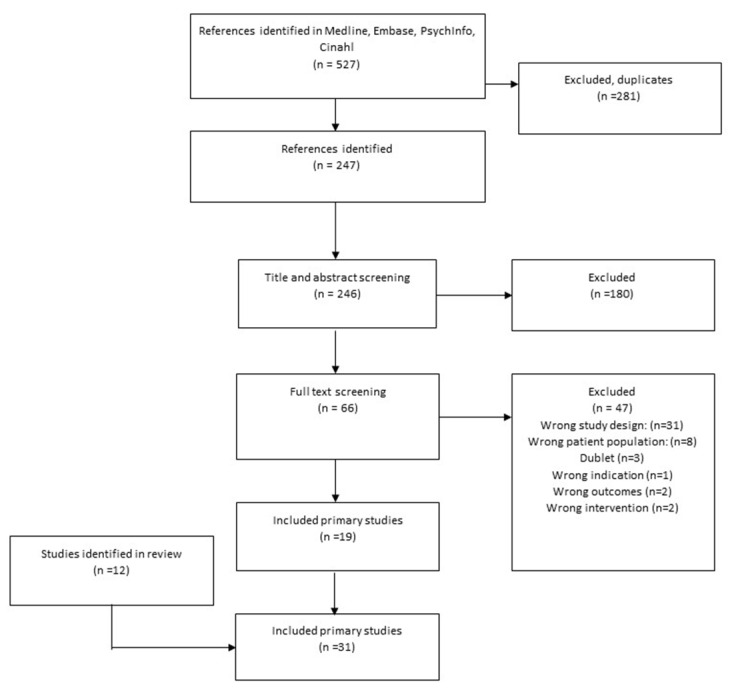
Flowchart of the search for primary studies in 2017 and 2020.

**Figure 3 nutrients-13-01226-f003:**
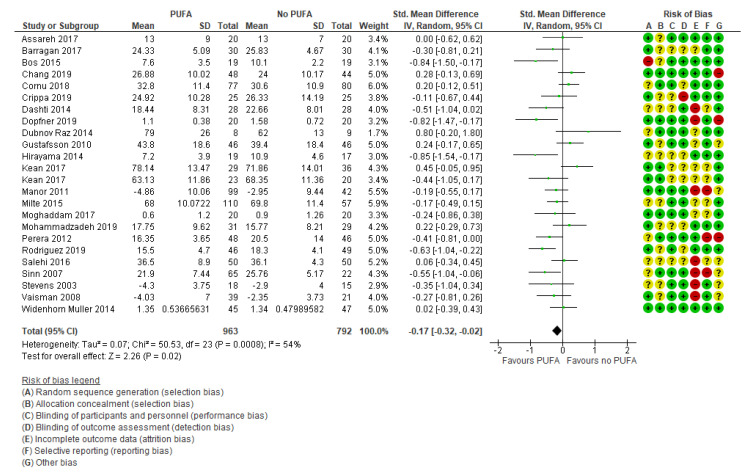
Forest plot of comparison: PUFA vs. placebo, outcome: parent-reported core symptoms (end of treatment). Green square indicates summary estimates of the individual studies. Black diamond indicates total summary effect estimate.

**Figure 4 nutrients-13-01226-f004:**
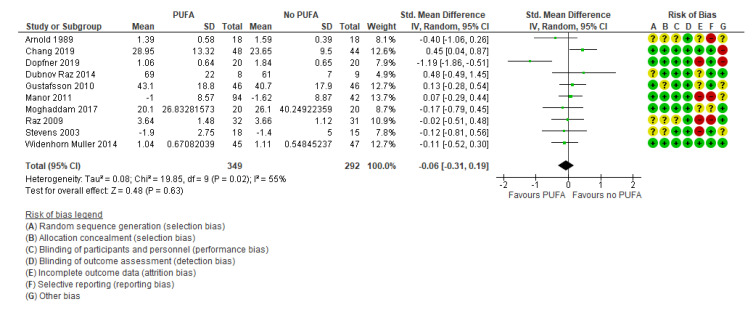
Forest plot of comparison: polyunsaturated fatty acids (PUFA) vs. placebo, outcome: teacher-reported core symptoms (end of treatment). Green square indicates summary estimates of the individual studies. Black diamond indicates total summary effect estimate.

**Figure 5 nutrients-13-01226-f005:**
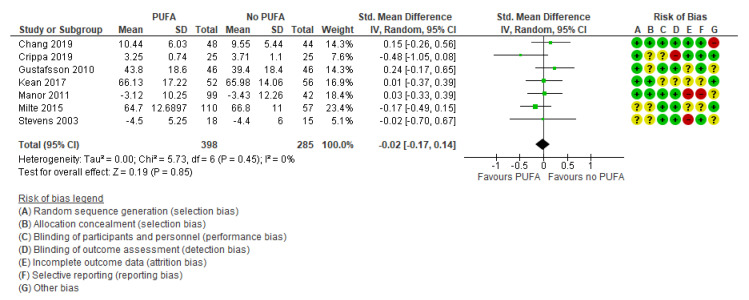
Forest plot of comparison: PUFA vs. placebo, outcome: parent-reported behavioral difficulties (end of treatment). Green square indicates summary estimates of the individual studies. Black diamond indicates total summary effect estimate.

**Figure 6 nutrients-13-01226-f006:**
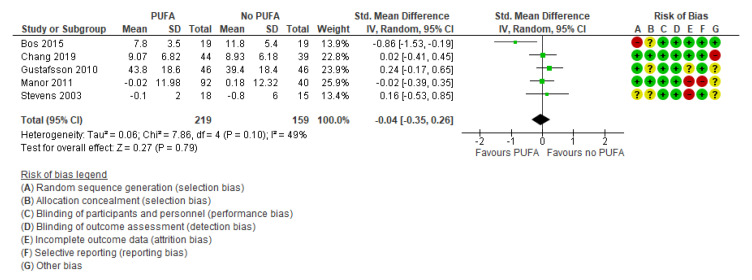
Forest plot of comparison: PUFA vs. placebo, outcome: teacher-reported behavioral difficulties (end of treatment). Green square indicates summary estimates of the individual studies. Black diamond indicates total summary effect estimate.

**Figure 7 nutrients-13-01226-f007:**
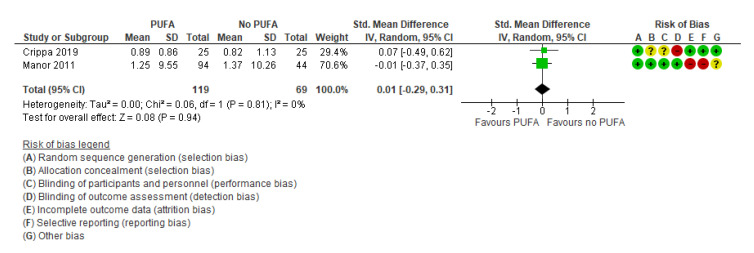
Forest plot of comparison: PUFA vs. placebo, outcome: quality of life (longest follow-up time (minimum 3 months after end of treatment)). Green square indicates summary estimates of the individual studies. Black diamond indicates total summary effect estimate.

**Figure 8 nutrients-13-01226-f008:**
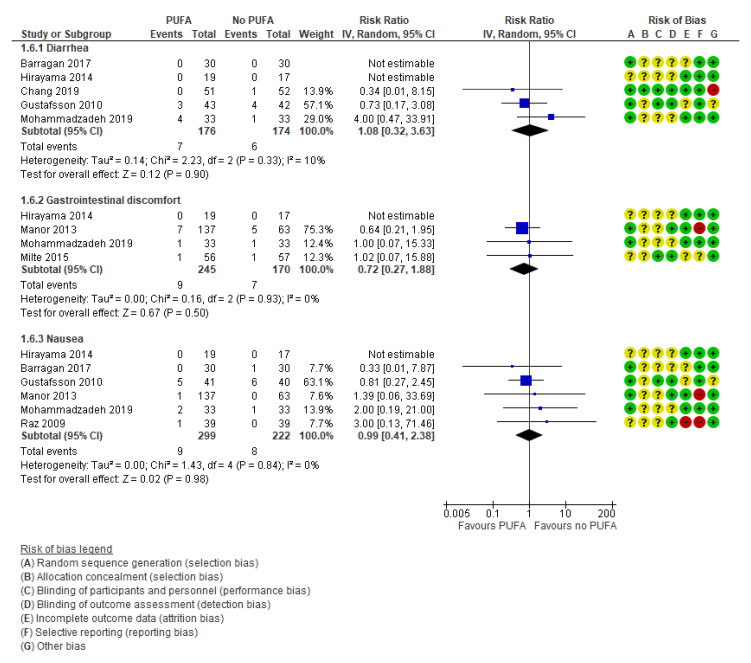
Forest plot of comparison: PUFA vs. placebo, outcome: side effects (end of treatment). Blue square indicates summary estimates of the individual studies. Black diamond indicates total summary effect estimate.

**Table 1 nutrients-13-01226-t001:** Population, Intervention, Comparison, and Outcome (PICO) criteria for inclusion and exclusion of studies.

Population	Children and adolescents between the age of 6 and 18 years (≥6 and ≤18), diagnosed with ADHD in accordance with ICD-10 or DSM criteria (both 4 and 5) for ADHD.
Intervention	Supplementation of polyunsaturated fatty acids (PUFAs). We included studies investigating both omega 3 and 6 fatty acids.
Comparison	No treatment—placebo and/or regular diet.
Outcome, primary	ADHD core symptoms, parent rated ADHD core symptoms, teacher rated Timing and effect measures ADHD core symptoms, both parent and teacher rated, were investigated at end of treatment. Minimal clinically important difference (MCID) 30% mean total score change difference between treatment groups, which is equivalent to between-treatment difference of 5.2 to 7.7 points [19,20,21].
Outcome, secondary	Behavioral difficulties, parent rated Behavioral difficulties, teacher rated Quality of life Diarrhea Gastrointestinal discomfort Nausea Timing and effect measures Behavioral difficulties both parent and teacher rated was investigated at end of treatment. Quality of life was investigated at the longest follow-up time (minimum 3 months after end of treatment). In our published protocol, we initially planned to assess diarrhea, gastrointestinal discomfort, and nausea at longest follow-up. This was later changed to end of treatment, as the identified studies did not provide any follow-up data on these outcomes.
Study design	All randomized controlled studies, with interventions matching the defined research question.

**Table 2 nutrients-13-01226-t002:** Example of the search strategy in Embase.

#	Searches
1	exp unsaturated fatty acid/
2	Diet therapy/ or diet supplementation/
3	exp Fish oil/
4	exp Carnitine/
5	((fatty adj1 acid*) or ((Polyunsaturated or poly-unsaturated or unsaturated) adj1 (fat or fatty)) or omega-3 or omega3 or omega 3 or omega-6 or omega6 or omega 6 or (docosahexaenoic adj acid*) or (eicosapentaenoic adj acid*) or (arachidonic adj acid)).ti,ab,kw.
6	((fish adj1 oil*) or cod liver oil* or lax oil* or tuna oil* or carnitine or Levocarnitine or “L Carnitine” or L-carnitine or bicarnitine).ti,ab,kw.
7	((diet* or food or nutrition) adj1 (therapy or supplement*)).ti,ab,kw.
8	or/1–7
9	exp Attention Deficit Disorder/
10	(ADHD or (hyperkinetic adj1 disorder*) or (Attention adj1 Deficit adj1 Disorder) or (attention-deficit adj1 disorder)).ti,ab,kw.
11	9 or 10
12	8 and 11
13	limit 12 to (randomized controlled trial or controlled clinical trial)
14	(((random* or cluster-random* or quasi-random* or control?ed or crossover or cross-over or blind* or mask*) adj4 (trial*1 or study or studies or analy*)) or rct).ti,ab,kw.
15	(placebo* or single-blind* or double-blind* or triple-blind*).ti,ab,kw.
16	((single or double or triple) adj2 (blind* or mask*)).ti,ab,kw.
17	((patient* or person* or participant* or population* or allocat* or assign*) adj3 random*).ti,ab,kw.
18	14 or 15 or 16 or 17
19	12 and 18
20	13 or 19
21	limit 20 to (yr = “2017–2020” and (english or danish or german or norwegian or swedish))

## Data Availability

Data are available at the Danish Health Authority website (www.sst.dk), accessed on 4 April 2021.

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
