# Peer review of "Efficacy and Safety of Polyunsaturated Fatty Acids Supplementation in the Treatment of Attention Deficit Hyperactivity Disorder (ADHD) in Children and Adolescents: A Systematic Review and Meta-Analysis of Clinical Trials"

_nutrients, 2021, doi:10.3390/nu13041226_

Round 1

Reviewer 1 Report

The paper entitled “Efficacy and safety of polyunsaturated fatty acids supplementation in the treatment of Attention Deficit Hyperactivity Disorder (ADHD) in children and adolescents: a systematic review and meta-analysis of clinical trials” focuses on a timely topic that fits well within Nutrients. Moreover, authors offer a relevant and interesting discussion about the clinical effects of using PUFA as a supplementation in the treatment of ADHD and whether it is or it is not supported by evidence. Findings may be of relevance for clinicians and physicians attending children with ADHD so they can provide evidence-informed intervention strategies to these children and their families. The systematic review and meta-analysis follow the PRISMA guidelines and recommendations and it is methodologically sound. However, there are some minor and mayor aspects that should be revised.

Mayor aspects

  1. What was the time period of published studies for the systematic review? As the search conducted in 2013 and then in 2017 is already published, it seems that only six studies were reviewed for the first time in the present paper. Authors should further empashize the justification to perform this update and the new contributions of the 2020 review.
  2. Authors mention that they followed the PICO structure, but they do not specify the PICO question nor they describe the specific search strategy. Moreover, recent recommendations promote the use of the PICOS structure, where the “S” refers to the Study design. Although the complete search protocol is fully described in Supplementary Materials, authors should include a description of the terms and an example of the search strategy used in at least one database. In addition, please develop the PICOS question in the Material and Methods section.

Minor aspects

  1. Abstract. Authors should include the protocol registration code in the abstract.
  2. Lines 40-41. It seems to be a typo there, as the sentence after “…the age of 18 years” is oddly written.
  3. There seems to be too many missing spaces between paragraphs. Additional “enters” to separate paragraphs or sections are not necessary.
  4. In Figure 2, please include the references identified through database searching (including all the databases included) and records after duplicates removed.

Author Response

RESPONSE TO REVIEWERS' COMMENTS

We thank the Editor and reviewers for their comments and suggestions which we believe have strengthened the manuscript further. We appreciate the opportunity to elaborate on the points raised by the reviewers.

Please find below our point-by-point replies to the reviewers’ comments, which are formatted in bold, followed by our reply in blue colored characters.

Revisions in the manuscript are clearly highlighted, using the "Track Changes" function in Microsoft Word, so that changes are easily visible to the editors and reviewers.

Reviewer 1:

Comments and Suggestions for Authors

The paper entitled “Efficacy and safety of polyunsaturated fatty acids supplementation in the treatment of Attention Deficit Hyperactivity Disorder (ADHD) in children and adolescents: a systematic review and meta-analysis of clinical trials” focuses on a timely topic that fits well within Nutrients. Moreover, authors offer a relevant and interesting discussion about the clinical effects of using PUFA as a supplementation in the treatment of ADHD and whether it is or it is not supported by evidence. Findings may be of relevance for clinicians and physicians attending children with ADHD so they can provide evidence-informed intervention strategies to these children and their families. The systematic review and meta-analysis follow the PRISMA guidelines and recommendations and it is methodologically sound. However, there are some minor and mayor aspects that should be revised.

Thank you for the comment

Mayor aspects

What was the time period of published studies for the systematic review? As the search conducted in 2013 and then in 2017 is already published, it seems that only six studies were reviewed for the first time in the present paper. Authors should further emphasize the justification to perform this update and the new contributions of the 2020 review.

Thank you for the comment.

The justification for the stepwise search strategy, is that the Danish Health Authority prioritize to enrich existing guidelines with all available literature approximately every third year. As stated in the results on line 214 and onward, the contribution divided by year was, as follow: 12 studies in 2013, additional 11 studies in 2017, and further 8 studies in 2020.

In retrospect, it seems that the rationale for updating the existing guideline every third year is strong, as this is a very productive research field. Thus, the contribution of the present review is the qualitative and quantitative synthesis of the combined evidence base to date.

We have highlighted the search flow in the manuscript, on line 144 and onward:

As a part of the conduction of the Danish national clinical guideline on ADHD by the Danish Health Authority, a literature search were performed by KB The literature search for this current review constituted an update of the search strategy initially performed in September 2013 (with no restriction on date), and again in November 2017 (dates restricted to 2013-2017), and again in June 2020 (dates restricted to 2017-2020 ) as a part of the con-duction of the Danish national clinical guideline on ADHD by the Danish Health Author-ity. The systematic literature search for this update, was performed in multiple databases including Medline, Embase, Cinahl, and PsycInfo in June 2020 for publication years 2017-2020 and limited to randomized controlled trials. The original search strategy in-cluded a search for both systematic reviews and individual randomized controlled trials. The justification for the stepwise search strategy, is that the Danish Health Authority prior-itize to enrich existing guidelines with all available eligible literature approximately every third year. All search strategies for the 2013, 2017 and 2020 searches included medical subject heading (MeSH) and text words related to our eligibility criteria.      

Authors mention that they followed the PICO structure, but they do not specify the PICO question, nor they describe the specific search strategy. Moreover, recent recommendations promote the use of the PICOS structure, where the “S” refers to the Study design.

Thank you for the suggestion. We fully agree and have highlighted the PICO question in both text on line 81 and added table 1 that specifies PICOS, including the rationale for selecting RCTs, only.

Specifically, the PICO question was whether children and adolescents aged 6-18 years with ADHD should be offered PUFA supplementation. The definition of population, intervention, comparator, outcomes and study design is specified in table 1. The research question is best addressed through a randomized study design, in order to evaluate the effectiveness of the intervention so that issues regarding both measured and unmeasured confounding are minimized.

Although the complete search protocol is fully described in Supplementary Materials, authors should include a description of the terms and an example of the search strategy used in at least one database.

We have included an example of the intervention search terms on line 110 and an example of an entire search strategy conducted in Embase  in table 2.

As an example MeSH terms for the intervention were “unsaturated fatty acid” or “Diet therapy” or “diet supplementation” or   “Fish oil” or “Carnitine “, or text words for the intervention ((fatty adj1 acid*) or ((Polyunsaturated or poly-unsaturated or unsaturated) adj1 (fat or fatty)) or omega-3 or omega3 or omega 3 or omega-6 or omega6 or omega 6 or (docosahexaenoic adj acid*) or (eicosapentaenoic adj acid*) or (arachidonic adj acid)) or ((fish adj1 oil*) or cod liver oil* or lax oil* or tuna oil* or carnitine or Levocarnitine or "L Carnitine" or L-carnitine or bicarnitine) or ((diet* or food or nutrition) adj1 (therapy or supplement*)) (for further details please see the example presented in Table 2 and in the search protocols provided in Supplementary Material). Both the original and updated searches were limited to literature written in English, Danish, Norwegian and Swedish, due to limitation in language proficiency in the author group. Moreover, to ensure that any relevant studies were not missed by the search, content experts from the guideline working group were conferred, and reference lists of included articles and previous reviews were screened for potentially relevant studies. Conference abstracts were considered, if data was not published elsewhere. Study authors were not contacted to identify additional studies.

In addition, please develop the PICOS question in the Material and Methods section.

Please, see the previous reply regarding PICOS.

Minor aspects

Abstract. Authors should include the protocol registration code in the abstract.

The last line in the abstract include the protocol registration code. We have now added more text to highlight that the reference to PROSPERO was part of the abstract on line 31:

The protocol is registered in PROSPERO ID: CRD42020158453  

Lines 40-41. It seems to be a typo there, as the sentence after “…the age of 18 years” is oddly written.

There seems to be too many missing spaces between paragraphs. Additional “enters” to separate paragraphs or sections are not necessary.

Thank you for the suggestion. The line on 40-41 has now been deleted, and Additional “enters” to separate paragraphs or sections have now been deleted throughout the manuscript.

In Figure 2, please include the references identified through database searching (including all the databases included) and records after duplicates removed.

Thank you for highlighting the short comings of figure 2, and we fully agree. Figure 2, now include the references identified through database searching (including all the databases included) and records after duplicates removed

Reviewer 2:

The authors conducted this systematic review with meta-analysis evaluating the benefits and risks of PUFA supplements on ADHD children, which followed the guidelines of the Cochrane Collaboration and PRISMA. After the analyses based on 31 relevant RCTs, the authors made the conclusion that the benefits and concerns of PUFA treatment are not supported by current low-quality evidence. The overall quality of this review is great. There is only one question need to be clarified by the authors:

In line 217, the authors stated that "no clinically relevant effect was found", while the SMD for the ADHD core symptoms is statistically significant. Authors should add more information about the definition of a clinically relevant effect in this topic.

According to Cochranes Handbook, the rules of thumb for interpreting SMDs is that a SMD of 0.2 represents a small effect (https://training.cochrane.org/handbook/current/chapter-15#section-15-5). The rating scales used to measure ADHD core symptoms varied greatly in the included studies, but on the Parent ADHD rating scale (range 0 to 54), minimal clinically important difference (MCID) has been defined as a 30% mean total score change difference between treatment groups, which is equivalent to between-treatment difference of 5.2 to 7.7 points. If we convert the SMD results from the meta-analysis into a MD using the standard deviation of the control group in the largest study that uses Parent ADHD rating scale (Cornu et al. 2018, n in control = 80), by means of the following formula (SMD (-0.17; 95% CI: -0.32, -0.02) x SD  in control group (10.9)), the MDcalculated = -1.85 (95% CI: -3.49, -0.22), which is equivalent to a decrease of 3.4% on the scale. Therefore, we do not assume that the results are clinically relevant. Information on minimum clinical important difference (MCID) has been added to table 1:

Minimal clinically important difference (MCID) 30% mean total score change difference between treatment groups, which is equivalent to between-treatment difference of 5.2 to 7.7 points [19-21].

The conversion of SMD to MD have been added to the results section on line 243:  

The results showed no clinically relevant effect was found on the primary outcome par-ent-rated ADHD core symptoms SMD: -0.17; 95% CI: -0.32, -0.02) (Figure 3), corresponding to a mean difference on the Parent ADHD rating scale of -1.85 (95% CI: -3.49, -0.22) calcu-lated from the endpoint SD in the control group of Cornu et al. 2018 [51]. The Parent ADHD rating scale ranges from 0 to 54, and thus the result is equivalent to a decrease of 3.4% on the scale (MCID is estimated to a decrease of 30% [19-21]).

Reviewer 2 Report

The authors conducted this systematic review with meta-analysis evaluating the benefits and risks of PUFA supplements on ADHD children, which followed the guidelines of the Cochrane Collaboration and PRISMA. After the analyses based on 31 relevant RCTs, the authors made the conclusion that the benefits and concerns of PUFA treatment are not supported by current low-quality evidence. The overall quality of this review is great. There is only one question need to be clarified by the authors:

In line 217, the authors stated that "no clinically relevant effect was found", while the SMD for the ADHD core symptoms is statistically significant. Authors should add more information about the definition of a clinically relevant effect in this topic.

Author Response

(The authors gave the same response as above.)

Round 2

Reviewer 1 Report

I would like to thank the authors, who have addressed all my comments. I think this paper will be a nice and relevant addition to Nutrients.